# Liquid Crystal-Tuned Planar Optics in Terahertz Range

Hongguan Yu, Huacai Wang, Qiguang Wang, Shijun Ge * and Wei Hu *

Key Laboratory of Intelligent Optical Sensing and Manipulation, College of Engineering and Applied Sciences, Nanjing University, Nanjing 210023, China
* Correspondence: geshijun@nju.edu.cn (S.G.); huwei@nju.edu.cn (W.H.)

**Abstract:** Recently, terahertz waves of higher frequencies compared to microwave and radio frequency have shown great potential in radar detection and high-speed wireless communication. To spatially control the wavefront of terahertz beams, various novel components, such as terahertz filters, polarization converters and lenses, have been investigated. Metamaterials and metasurfaces have become the most promising technique for the free manipulation of terahertz waves. Metadevices integrated with liquid crystals have been widely used in active terahertz devices. In this review, the birefringence of liquid crystals in the terahertz band and terahertz devices based on liquid crystals are summarized. By integrating liquid crystals with plasmonic metamaterials, the functions become dynamically adjustable and are reconstructed. Utilizing liquid crystals to change the resonance of metamaterials, tunable filters, absorbers, and programmable metasurfaces are realized. To solve the problem of low efficiency, terahertz wavefront shaping devices based on dielectric metasurfaces and liquid crystals, such as a variable deflection angle grating and zoom metalenses, are presented. Finally, we discuss and anticipate the future developments of liquid-crystal-integrated meta-devices, which will inspire broad applications in terahertz communication and imaging.

**Keywords:** liquid crystal; metamaterial; tunable; terahertz





## 1. Introduction

The explosive expansion of data traffic in wireless communication systems prompts the use of spectra with higher carrier frequencies to increase the bandwidth and provide larger capacity. The terahertz (THz) spectrum, between microwave and infrared waves (frequency range 0.1 to 10 THz), theoretically supports a communication capacity of terabits per second (Tbps) [1]. THz is considered the most promising technology for next-generation wireless communication, and is known as "the last piece of the RF spectrum puzzle for communication systems" [2]. THz waves can penetrate most nonmetallic and nonpolar materials, such as cloth, paper, and wood, making them very suitable for nondestructive detection and security checks [3,4]. Currently, one of the biggest obstacles to THz communication is the absorption of water vapor in the atmosphere which makes it difficult to transmit high fidelity over long distances [5]. Therefore, it is necessary to develop highly efficient and compact passive and active devices, such as focusing [6], deflecting [7] and filtering [8] devices, to control the propagation of THz beams in free space.

Compared with electrical or optical counterparts, THz devices are far from mature. THz exceeds the cutoff frequencies of many semiconductor amplifiers and mixers, making them incompatible with existing RF components [9]. High-speed amplitude and phase coding based on photonics provide an opportunity for high-efficiency THz modulations [10,11]. Early refracted or diffracted THz devices are made of polymers or crystals, and their bulky sizes and fixed functions limit the miniaturization, integration, and versatility of THz apparatuses. Recently, metamaterials and metasurfaces based on subwavelength artificial electromagnetic microstructures have become excellent means for THz wavefront shaping [12–14]. Research on metamaterials began in 1968, when Veselago first proposed a medium with negative permeability [15]. In 1996, negative effective permeability was

experimentally demonstrated by significantly enhanced magnetic resonance using artificial structures with periodic split ring resonators (SRRs) [16]. Early metamaterials were used as frequency selective filters [17]. With the development of micro/nano processing and electromagnetic simulations, research on artificial electromagnetic microstructures has been extended to the THz band [18]. By properly designing the unit structures, various filters [19], polarization converters [20], lenses [21] and other components have been achieved. The modulation of electromagnetic waves by metamaterials mainly depends on the surface plasmon resonance (SPR) of the metallic microstructure, which is usually narrowband and has high backscattering loss and ohmic loss, especially in the THz band [22]. To improve efficiency, all-dielectric metasurfaces [23] and multilayer metal metamaterials [24] have been adopted.

Metamaterials are compact, flexible in function, and compatible with CMOS technology, but their functions are usually fixed once fabricated. Active tunable THz elements are realized by integration with various functional materials [25]. The dielectric properties of these functional materials change under the excitation of an external field, thus regulating the electromagnetic properties of metamaterials accordingly. Compared with other functional materials, liquid crystals (LCs) exhibit the superiorities of mature fabrication and low processing cost [26]. Driven by external stimuli such as electric fields, temperature and magnetic fields, the birefringence of LCs which covers a broadband range from ultraviolet to microwave, can be tuned [27]. Therefore, LC-based and LC-integrated THz devices have attracted intensive attention in recent years. With the aid of artificial intelligence algorithms, LC-integrated metadevices promote the dynamic control of THz beams and are diversely applied in THz radar detection and compressed sensing imaging. However, the lack of systematic review of the recent progress in this field hinders efforts to rapidly meet the requirements of LC-integrated THz devices.

We systematically review the recent progress in LC-based devices and LC-integrated metadevices in the THz band. In the second section, we present the developments of LC materials in the THz band and the progress in LC-based THz devices. In the third section, we review LC-integrated plasmonic metamaterials with a focus on various filters, spatial THz wave modulators (STM) and programmable metasurfaces. In the fourth section, LC-tuned all-dielectric metasurfaces and their applications in the dynamic focusing and deflection of THz beams are presented. Finally, we propose the future development of LC-tuned planar THz optics. This review presents the development of LC-based THz wave modulation and paves the way for compact, versatile and intelligent THz components for communication, imaging, and detection.

## 2. Properties of LCs and LC-Based THz Devices

Known for their dominant role in information displays, LCs, featuring both fluidity and anisotropy, can also be adopted for dynamic THz wave manipulations. The birefringence, $\Delta n$, is a key parameter of the LC which determines the function of wavefront regulation. For instance, LC THz modulators work on accumulated phase modulations, which are related to the birefringence and thickness of LCs. Thick cells suffer from slow responses and poor alignments; thus, large birefringence is in high demand. The properties of 5CB at 0.3–1.4 THz were measured 20 years ago, with a $\Delta n$ of 0.21 [28]. E7 exhibits a birefringence range of 0.130–0.148 at 0.2–2.0 THz [29]. In 2012, the birefringence of 18-series LCs was tested, and that of 1825 reached 0.36 at 0.7–3 THz. Moreover, the absorption of these LCs was negligible in the experiments [30,31]. In 2012, NJU-LDn-4 was synthesized, exhibited an average birefringence of 0.306 in 0.4–1.6 THz and kept the nematic phase in a large temperature range covering room temperature [32]. The nanoparticles doped E7 LC leads to a 10% increase in birefringence in 0.3–3 THz [33]. Table 1 shows the birefringence of these LC materials used in the THz band.

**Table 1.** Birefringence of LCs in the THz band.

| | Freq (THz) | $n_o$ | $n_e$ | $\Delta n$ | Temp (K) |
|---|---|---|---|---|---|
| 5CB [28] | 0.3–1.4 | 1.59–1.83 | 1.74–2.04 | 0.13–0.21 | 298 |
| E7 [29] | 0.2–2.0 | 1.557–1.581 | 1.690–1.704 | 0.130–0.148 | 299 |
| 1825 [30] | 1.5 | 1.574 | 1.951 | 0.377 | Room temperature |
| BL037 [34] | 0.2–2.0 | 1.58–1.64 | 1.78–1.80 | 0.17–0.20 | 294 |
| RDP-97304 [34] | 0.2–2.0 | 1.55–1.61 | 1.77–1.79 | 0.18–0.22 | 294 |
| NJU-LDn-4 [32] | 0.4–1.6 | 1.5–1.51 | 1.80–1.82 | ~0.31 | Room temperature |

As a typical element, the filter plays an important role in the terahertz band. The birefringence of LC is adopted for terahertz filters. Pan's group produced Lyot filters based on the combination of a magnetic field-controlled parallel-oriented LC cell exhibiting a fixed phase delay, and vertically oriented LC cells with an adjustable phase delay. Through changing the direction of the magnetic field, a modulation range of 40% at 0.388–0.564 THz was demonstrated [35]. Later, magnetically tuned Solc filters were made using two cells with vertically oriented LCs. The modulation bandwidths were 0.176–0.293 THz and 0.474–0.794 THz, respectively. The corresponding modulation ranges were 66.5% and 67.5%, separately [36]. A 60 GHz band stop filter at approximately 0.3 THz, as shown in Figure 1a,b, was realized by electrically controlling multiple LC layers [37]. A notch filter with a bandwidth of 0.35–0.7 THz was realized through electrically switching a single-layer LC cell [38]. THz filters based on LC-filled photonic crystals were theoretically investigated as well [39]. In these early works, almost all of them suffered from thick LC layers slowing down the response and increasing the absorption loss. The stacking of multiple LC cells reduces the thickness of a single cell but induces a more complicated configuration and extra interface loss.

The birefringence of the LC is also adopted for phase shifters. Through electrically driven nematic LC [40,41], shown in Figure 1c, polymer-stabilized LC [42] and hydrogen-bonded LC [43], different THz phase shifters were realized. THz LC wave plates were demonstrated in a similar way. A broadband tunable THz wave plate made of NJU-LDn-4 was reported [44]. In this element, metal wire grids and a few layers of graphene were used as conductive electrodes, illustrated in Figure 1d. Due to the large birefringence of NJU-LDn-4 in the THz band, a broadband quarter wave plate and a half wave plate were produced. By further introducing a double-layer structure, the bandwidth can be further expanded to 0.5–2.5 THz for a quarter wave plate. Compared to the transmitted THz LC wave plates, the reflective ones require only 50% LC to achieve the same modulation, which accelerates the modulation and reduces the insertion loss [45]. Chang's group used a 600 μm dual-frequency LC (DFLC) to form a cell whose birefringence is frequency dependent. Via tuning the frequency from 1 kHz to 100 kHz under a constant electric field, a tunable quarter wave plate above 0.68 THz and a half wave plate above 1.33 THz were realized, as shown in Figure 1e [46].

Besides the above devices, LCs have been utilized in abundant THz applications. The THz beam was deflected by an electrically controlled LC wedge cell, as seen in Figure 2a [47]. By injecting E7 into the groove of a silicon grating, as seen in Figure 2b, the intensity ratio between the 0th order and 1st order was magnetically tuned from 4:1 to 1:2 at 0.3 THz [48]. The twisted nematic LC cell accomplished the linear polarization conversion of the incident THz beam, and applying an electric field further adjusted the polarization [49]. Polarization-dependent imaging was achieved by inserting a uniformly oriented LC polymer film with different orientations compared to the orthogonal polarizers. The output intensity was varied by adjusting the orientations, which can be used as attenuators or switches [50]. In previous works, the LC alignments were either controlled by a strong magnetic/electric field or by mechanical rubbing. Both of them were restricted to homogeneous alignment. The appearance of photoalignment [51] makes the micropatterning of LC achievable. Our group used the photoalignment technique to encode the geometric phase into LC cells. By

this means, q-plates [52] and bifocal lenses [53] (shown in Figure 2c) were demonstrated. Similar procedures were carried out in LC polymers as well; thus, lenses, deflectors and axicons were realized, as illustrated in Figure 2d [54]. These devices eliminate the substrates of traditional THz LC devices, making them more compact and energy efficient. The Brewster angle was introduced by the refractive index difference between the silicon and the LC. In the range of 0.2–1.6 THz, a 99.6% modulation depth was achieved based on a double-switch electrical modulation. Linear to circular polarization conversion was achieved in 0.4–1.8 THz, as seen in Figure 2e. This design modulates both the intensity and the phase of the THz wave simultaneously, which is a milestone in THz wave manipulation [55].

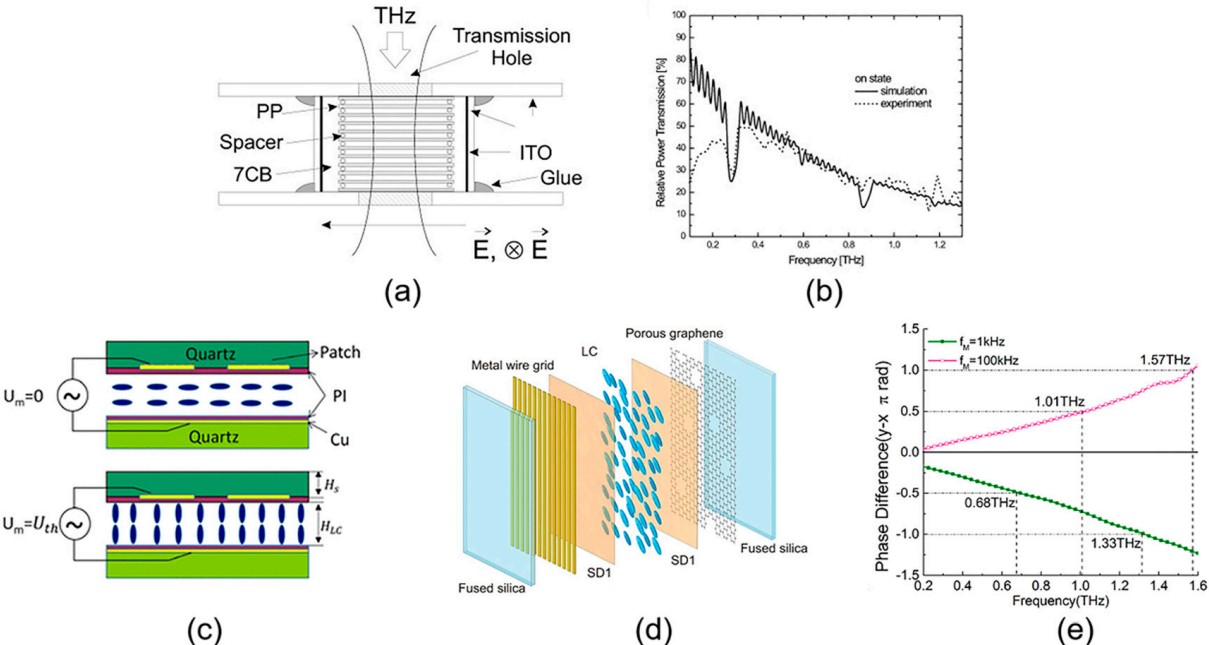

**Figure 1.** (**a**) Illustration of the stop-band filter and (**b**) corresponding transmission at the ON state [37]. (**c**) Schematic of the electrically driven phase shifter [41]. (**d**) Configuration of the broadband LC THz wave plate [44]. (**e**) The dependency of the phase difference on the frequency of the DFLC THz wave plate [46].

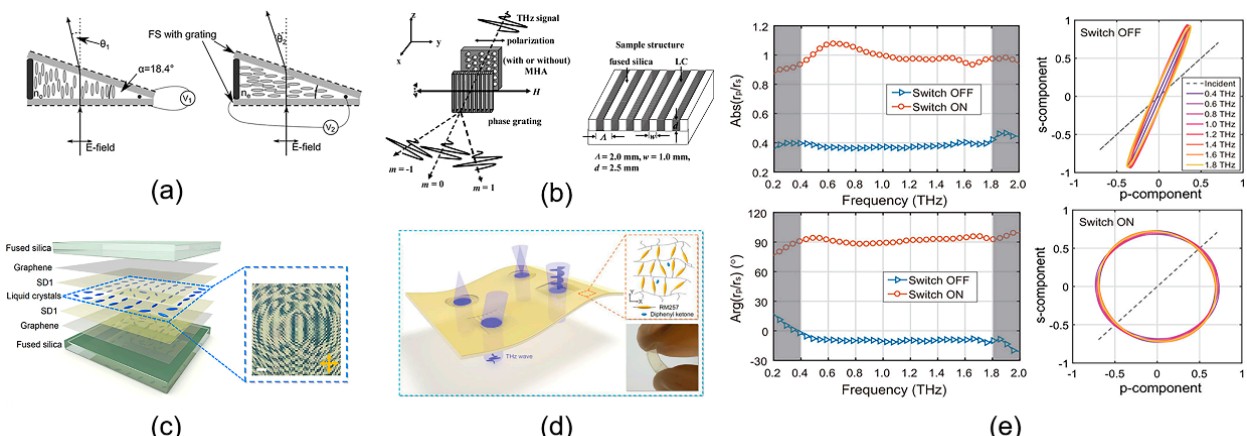

**Figure 2.** Schematic diagrams of (**a**) the wedge cell [47] and (**b**) the LC phase grating [48]. (**c**) Structure and micrograph of a bifocal lens. Scale bar: 1 mm. [53] (**d**) Photopatterned LC polymer-based planar THz optics [54]. (**e**) The magnitude ratio (**top-left**) and phase difference (**bottom-left**) between the p and s components and polarization output at the OFF (**top-right**) and ON (**bottom-right**) states [55].

## 3. LC-Integrated Plasmonic Metadevices

Plasmonic metamaterials consist of artificial metallic micro/nanostructures. By tailoring the structures and shapes of metamaterial units, functions not available with natural materials, such as [56], invisible cloaks [57], and perfect absorption [58], can be achieved. To realize dynamic or switchable functions, metamaterials are usually combined with semiconductors [59], graphene [60], vanadium dioxide [61], and other functional materials. LCs have attracted special attention due to their mature and cost-efficient fabrication [62]. However, they still suffer from slow operating speed as well as limited modulation depth, and there is still a long way toward practical active metadevices.

In recent years, a lot of THz absorbers, phase shifters and polarization converters based on LC-integrated metamaterials have been presented. Because of their applications in sensing and imaging fields, THz absorbers have been widely concerned [63–65]. In 2011, a tunable bandpass filter was theoretically designed, which uses a woodpile metallic photonic crystal as a resonator and electrode and LCs as a defect layer to fill the photonic crystal [66]. In 2013, Shrekenhamer et al. demonstrated LCs embedded into metamaterials to form metal–LC–metal absorbers, and 30% amplitude modulation of absorption at 2.62 THz and over 4% bandwidth adjustment were presented, as shown in Figure 3a [67]. By combining different geometries of plasmonic resonators, a multiband tunable metamaterial absorber can be realized [68]. In Figure 3b, a few layers of porous graphene are integrated into the metamaterial to apply a uniform electric field to LCs and achieved an amplitude modulation of ~80% at a voltage of 10 V [69]. In addition to metamaterial absorbers, modulators are also used to modulate THz wave amplitudes. Different THz modulators based on LC-integrated metamaterials have been reported. However, most of these devices operate in reflection mode, limiting their applications in THz systems. As shown in Figure 3c, Yang et al. reported a metadevice that worked in transmission, with high modulation depth and low insertion loss, by integrating two layers of metamaterials with LCs [70]. In 2018, an LC-integrated metadevice operating in transflective mode was reported, and its structure is shown in Figure 3d. The comb electrodes exhibited polarization selectivity and realized a dynamic electromagnetically induced transparency analog (EIT) in transmission mode, and a dynamic absorber in reflection mode [71].

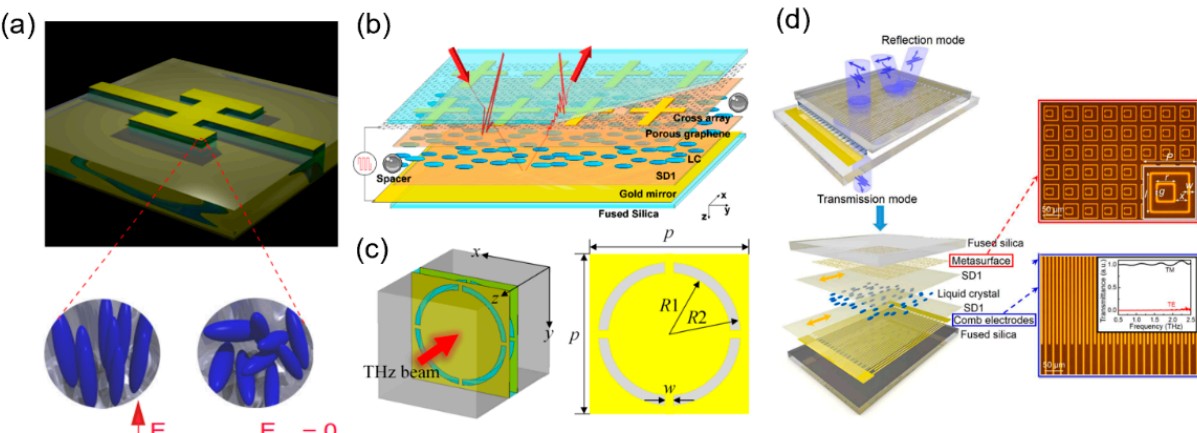

**Figure 3.** (**a**) Schematic diagram of the metal–LC–metal absorber [64]. (**b**) Illustration of a few-layer porous graphene-assisted metamaterial absorber [66]. (**c**) Structure of a double layer plasmonic metamaterial THz modulator [67]. (**d**) Structure of a THz filter with dual operating modes [68].

In the above tunable terahertz metadevices, LCs act as the intermedium with a tunable refractive index. Through reorienting the LCs driven by an external field, dynamic modulation of the THz intensity and phase can be realized. In this way, the modulation speed reaches the millisecond scale, but the modulation depth is commonly low. By integrating LCs as tunable waveplates with polarization-dependent metamaterials, dynamic THz resonant switches with large modulation depths of THz amplitude and phase were achieved,

as shown in Figure 4a [72]. Through thermally [73], photo [74], or magnetically [75] tuning the LC birefringence, the resonances of metamaterials were also varied. In Figure 4b, an optically tunable and thermally erasable THz modulator was realized by doping LC with the azo dye methyl red [76]. Based on the rewritable photo-patterning of SD1, a spatial terahertz modulator featured by photo-reconfigurable and electric-switchable is also designed [77]. In addition to traditional nematic LCs, many other kinds of LCs have also been studied. Figure 4c shows that the response of the modulator can be significantly improved by integrating the in-plane-switching dual-frequency LC into a plasmonic metasurface [78]. As shown in Figure 4d, in 2018, Yin et al. proposed an integrated polymer network LC-based absorber, which achieved an electrically driven peak shift in 10 ms [79].

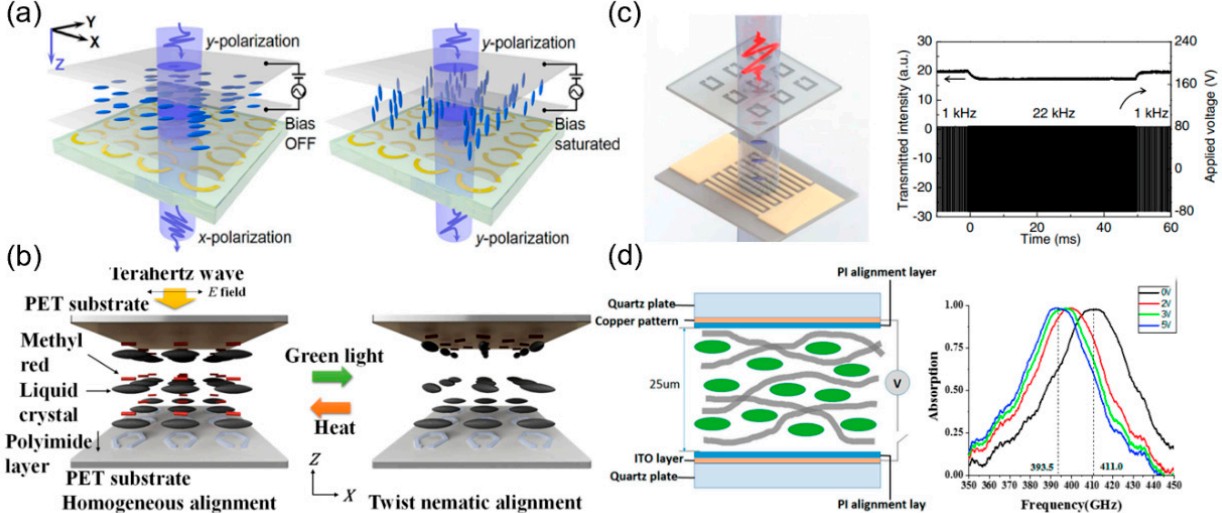

**Figure 4.** (**a**) Schematic diagram of an LC THz-resonant switch [72]. (**b**) Illustration of an optically tunable and thermally erasable THz modulator with LC doped by methyl red [76]. (**c**) Fast-response THz modulator fabricated by in-plane switching of dual-frequency LCs (**left**) and measured time-dependent transmission (**right**) [78]. (**d**) Fast-tunable THz absorber fabricated by polymer network LCs (**left**) and the measured absorptions (**right**) [79].

For LC-only THz devices, the required thickness of the LC layer is on the submillimeter scale, inducing a high driving voltage and slow response. The integration of metamaterials with LCs can significantly improve the response while maintaining a large phase shift. As shown in Figure 5a, Buchnev et al. achieved an amplitude modulation of 20% and a phase modulation of 40° by hybridizing 12 μm thick LCs with metasurfaces [80]. In 2019, Sasaki et al. designed a polarization converter using an LC/metal grid configuration, reducing the LC thickness by two orders of magnitude compared to pure LC-based devices [81]. In Figure 5b, benefiting from the wide-band characteristics of the metal gratings, broadband linear polarization generation can be achieved by mixing LCs as a refractive index variable environmental medium to actively tune the Fabry-Perot-like resonance in two orthogonally arranged metal gratings [82]. Figure 5c shows that the combination of LCs and chiral metamaterials enabled spin conversion and electronically manipulated optical chirality [83]. With the rapid development of THz technology, many other LC-integrated THz metadevices have been designed and demonstrated, such as the THz demultiplexer [84] and the spin controllable THz source [85], and so on.

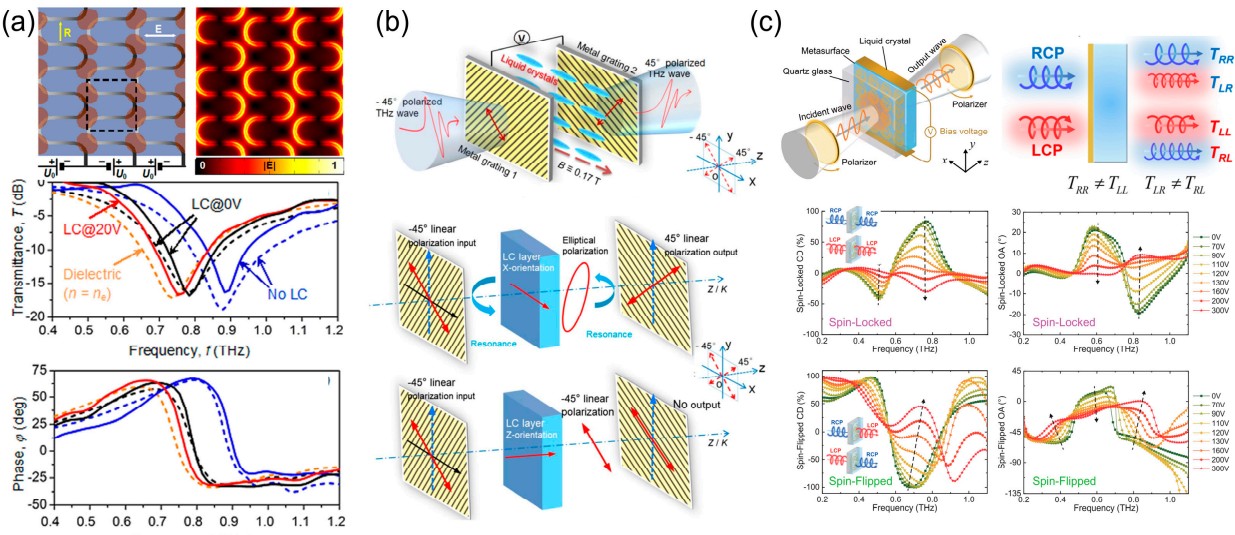

**Figure 5.** (**a**) Terahertz amplitude and phase modulations based on the fishnet metamaterial. The middle and bottom spectra present the dependencies of transmittance and phase on frequency at the OFF (solid black line) and ON (solid red line) states [80]. (**b**) Illustration of the polarization converter using an LC/metal grid. The output polarizations in the ON (**middle**) and OFF (**bottom**) states are presented [82]. (**c**) Spin-locked and spin-flipped manipulation by the LC-integrated chiral metamaterial. The middle and bottom figures show the circular dichroism (CD) spectrum and optical activity (OA) of the spin-locked and spin-flipped manipulation [83].

Dynamic THz beam steering has great potential in high-speed wireless communication, high-resolution imaging, and radar. Traditional radar relies on mechanical scanning for continuous beam steering to capture the target [86]. Phased arrays are considered good candidates for beam steering in the microwave band [87]. However, in the THz band, the high loss of the semiconductor switch restricts the phase shift. In 2014, Cui et al. proposed a programmable metasurface with the phase of each pixel unit switchable between 0 and π. Therefore, digitized dynamic manipulation of electromagnetic waves was realized by programming the coding sequences [88]. Illustrated in Figure 6a, in 2020, Wu et al. designed a programmable LC-integrated metasurface by coding the "0/1" phase to dynamically control beam steering, and the maximum deflection angle reached 32° [89]. In Figure 6b, Buchnev et al. adopted the LC sandwiched by S-type metamaterials to demonstrate an efficient and ultrathin spatial phase modulator [90]. By applying a gradient voltage, continuous phase retardation and high-resolution spatial phase control were achieved. To extend the range of phase modulation, Cui et al. used two layers of asymmetric metamaterials to spatially manipulate the transmitted THz beam, and demonstrated double-layer beam steering with a maximum deflection angle of 30°, as shown in Figure 6c [91]. The above designed programmable metasurfaces are all based on binary phase coding. To suppress unnecessary high-order diffraction, Xu et al. proposed a multibit coding scheme based on resonant switching and achieved maximum single-beam scanning of ±21° with significantly improved efficiency, as illustrated in Figure 6d [92]. STMs can spatially address THz amplitudes by binary coding each pixel as well. In 2014, Savo et al. designed a STM with 8 × 8 pixels using metamaterial absorbers. Each pixel was independently controlled by external FPGAs, and the average modulation depth of the reflectivity reached 75% [93]. Recently, Hu et al. designed a transflective STM by integrating LCs with a Fano-resonant metasurface and pixelated interdigital electrodes, as shown in Figure 6e. The modulation depth of each pixel reached 38.8% and 61.1% for the transmissive and reflective modes, respectively [94]. STMs also play important roles in THz imaging. In Figure 6f, Li et al. demonstrated an 8 × 8 pixelated STM using a dual-frequency LC integrated metasurface

absorber, and developed an auto-calibrated compressive sensing algorithm for THz single-pixel imaging [95].

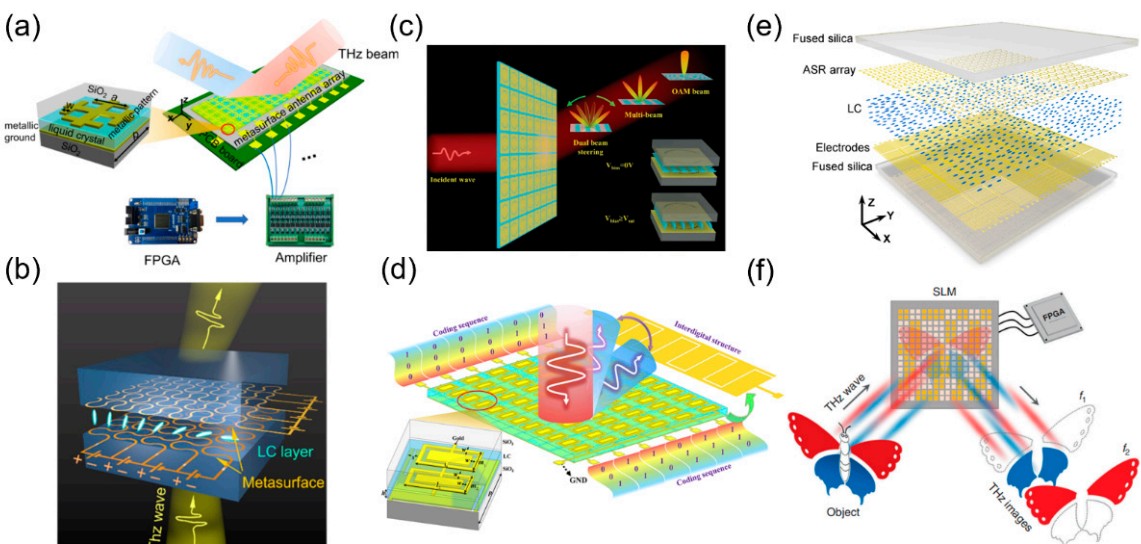

**Figure 6.** Schematic diagrams of (**a**) a THz programmable metasurface [89] and (**b**) an LC spatial phase modulator [90]. (**c**) Transmissive programmable metasurface composed of two layers of Fano metamaterial sandwiched by LCs [91]. (**d**) Resonant switch-based multibit coding [92]. Schematic diagrams of (**e**) transflective spatial THz wave modulator [94] and (**f**) dual-color spatial THz wave modulators for single-pixel imaging [95].

## 4. Tunable Dielectric Metasurfaces Based on LCs

For metallic metamaterials, a phase change from 0 to $2\pi$ cannot be achieved by changing the resonant frequency. Moreover, the low forward-scattering discounts the efficiency of wavefront modulation. Therefore, researchers introduced the concept of a metasurface into the THz band and designed a THz dielectric metasurface. Similar to glass in visible light, high-impedance silicon is transparent in the THz band. Additionally, silicon is perfectly compatible with existing nanofabrication technology, making it an ideal candidate for THz planar optics. By coding the phase of the THz wavefront, functions such as focusing, deflection and vortex beam generation can be realized. By appropriately designing the unit structure of dielectric metasurfaces, a subwavelength phase gradient can be realized in the two-dimensional plane. The phase introduced by dielectric metasurfaces is generally based on two different principles. One is the resonance phase related to the structural parameters of the unit. Dielectric meta-atoms interact with incident THz waves to excite Mie resonance and add a certain phase to the incident wave [96,97]. The resonant phase is expressed as [98]:

$$\varphi_R = \frac{2\pi}{\lambda} n_{eff} d \tag{1}$$

$\lambda$ is the operating wavelength, $n_{eff}$ is the effective refractive index of the dielectric silicon unit which is determined by the lengths and widths of meta-atoms, and $d$ is the height of the dielectric silicon unit. Due to the coupling of spin angular momentum and orbital angular momentum, the metasurface also produces geometric phase effects on incident electromagnetic waves. The geometric phase, also known as the Pancharatham–Berry phase (PB phase), is related to the evolution of the polarization state in space propagation and only depends on the spatial distribution of the optical axis of the structural unit. Assuming that the azimuthal angle of the optical axis is $\alpha$, its Jones matrix is [99]:

$$T(r, \varphi) = \cos\left(\frac{\delta}{2}\right)\begin{pmatrix} 1 & 0 \\ 0 & 1 \end{pmatrix} + i \sin\left(\frac{\delta}{2}\right)\begin{pmatrix} \cos 2\alpha & \sin 2\alpha \\ \sin 2\alpha & -\cos 2\alpha \end{pmatrix} \tag{2}$$

where $\delta$ is the retardation phase of the structure. When the left/right circularly polarized light incident $E_{in}(r, \varphi) = E_0(r, \varphi) \cdot (1, i\sigma)^T$, $\sigma = +1$ for left and $\sigma = -1$ for right, the output beam can be expressed as:

$$E_{out}(r, \varphi) = T(r, \varphi)E_{in}(r, \varphi) = E_0 \cos\left(\frac{\delta}{2}\right)\begin{pmatrix} 1 \\ i\sigma \end{pmatrix} + iE_0 \sin\left(\frac{\delta}{2}\right) \exp(i2\alpha\sigma)\begin{pmatrix} 1 \\ -i\sigma \end{pmatrix} \quad (3)$$

This indicates that the chirality of the incident light will be reversed and carries a phase delay of $\pm 2\alpha$ accordingly.

In the THz band, silicon-based metasurfaces are often fabricated by photolithography and deep reactive ion etching (DRIE). The manufacture is matured and available for mass production. As the function of the dielectric metasurface is strictly bound to the structure, it is totally fixed once the structure is fabricated. Methods for the realization of a tunable THz dielectric metasurface have become an important research direction. To introduce tunability to the THz dielectric metasurface, LCs are integrated. The tunability is divided into two species. One uses LCs as a birefringent medium to provide a tunable anisotropic refractive index. It allows LCs to be used as a tunable wave plate or a special environmental medium. In 2018, Hu et al. combined large birefringent NJU-LDn-4 NLCs in the THz band, with a circular silicon column array metasurface to fabricate a THz absorber, as shown in Figure 7a,c, with an adjustable power absorption rate and resonant peak position and which has a 47% modulation depth at 0.79 THz. Figure 7b shows that the absorption rate can reach approximately 80% at 0.8 THz when an electric field is applied. [100] However, there is no high absorption at the same frequency when bias is OFF. In 2020, Chang et al. made use of the controllable anisotropy of DFLCs. They combined DFLCs with a silicon column array metasurface to fabricate a wide-band THz modulator that can manually control its achromaticity and birefringence. This device can be turned from OFF state (no polarization conversion effect) to a controllable waveplate, exhibiting over 97% polarization conversion ratio (PCR) in 0.97–1.3 THz as a half-wave plate, or over 96% PCR in 0.67–1.3 THz as a quarter-wave plate [101]. In addition, they used polymer-dispersed liquid crystals (PDLCs) as the birefringent medium in another work. Driven by an electric field, the orientations of LCs rotate in the droplets of PDLCs, which in turn causes anisotropy variation and a reduction in dispersion in a wider bandwidth compared to the original silicon gradient grating metasurface [102]. In the same year, Hu et al. designed a linear polarization-multiplexing dielectric bifocal metalens, in which two orthogonal linear polarization incident lights have two different foci. They introduced LCs to select the outgoing light of different polarizations to achieve the bifocal function [103]. In 2021, Hu et al. used LCs as the medium of silicon metalens to control the environmental refractive index, thus changing the focal length of the metalens between 8.3 mm and 10.5 mm [104]. In 2022, Chang et al. fabricated a silicon-based elliptical-cylinder-array metasurface whose structures are shown in Figure 8a,b. As shown in Figure 8c, the purpose of manually controlling the anisotropy of THz devices was achieved through the dynamic coupling between the anisotropy caused by this metasurface in the THz band and the anisotropy caused by tunable LC orientations under different electric fields. With different initial LCs orientations, the difference between the $E_x$. and $E_y$ changes differently when applying the external electric fields, which is shown in Figure 8d,e. The highest circular dichroism reached 30 dB when comparing the power of two orthogonal electric fields [105]. Thus, using LCs as a tunable birefringence medium, one can change the working states of a dielectric metasurface.

The other way is to adopt LCs to supply an extra PB phase modulation. Moreover, the phase modulation can be temporarily erased by an external field. This gives the dielectric THz metasurface another method for tunability. In 2020, Hu et al. combined the PB phase of LC with the resonant phase of the silicon metalens (shown in Figure 9a), enabling a switch between an achromatic lens (Figure 9b,c) and a large chromatic lens (Figure 9d,e). This introduced a new approach for the combination of LCs and metasurfaces in the THz band, which has great potential in THz imaging and communications [106]. By orienting the LCs

into a PB phase grating pattern, an electrically adjustable focus separation between incident LCP waves and RCP waves for an off-axis bifocal THz metalens was achieved [103]. In 2022, Chang et al. fabricated a silicon-based PB phase metasurface and deflected incident light with different chirality in different directions, which realizes a dynamic modulation depth of >94% and maximum efficiency of over 50% for the different spin states. At the same time, the PB phase of the LCs is used to change the deflection direction of light under the same chirality incident light (as shown in Figure 10a–d), and a magnetic field is introduced to control the orientation of the LCs, making subsequent communication applications possible. As shown in Figure 10e,f, the deflection angle of about 0.9~1.1 THz changes its sign when the magnetic field is applied [107]. The tunable PB phase of LCs gives the metasurface tunable phase compensation and promotes advances in LC-integrated metadevices.

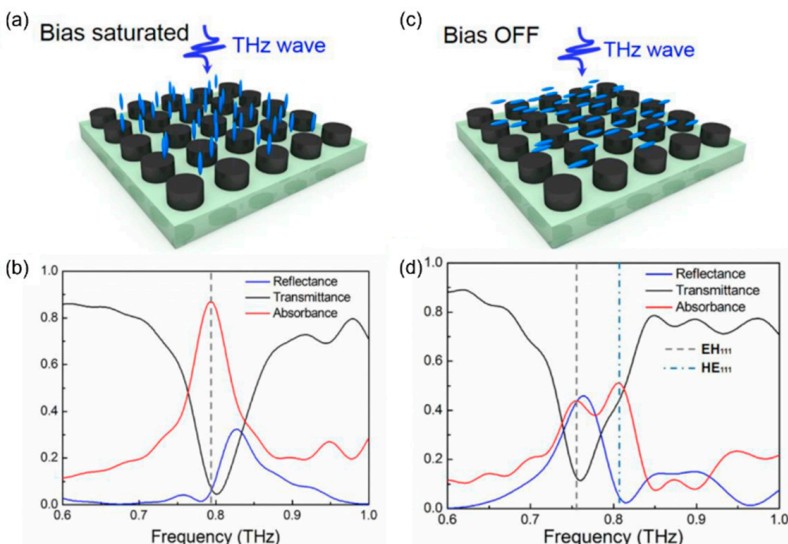

**Figure 7.** (**a**) The LC orientation in the bias-saturated state. (**b**) Simulated spectra of reflectance, transmittance, and absorbance in the bias-saturated state. (**c**) The LC orientation at the bias OFF state. (**d**) Simulated spectra of reflectance, transmittance, and absorbance in the bias OFF state [100].

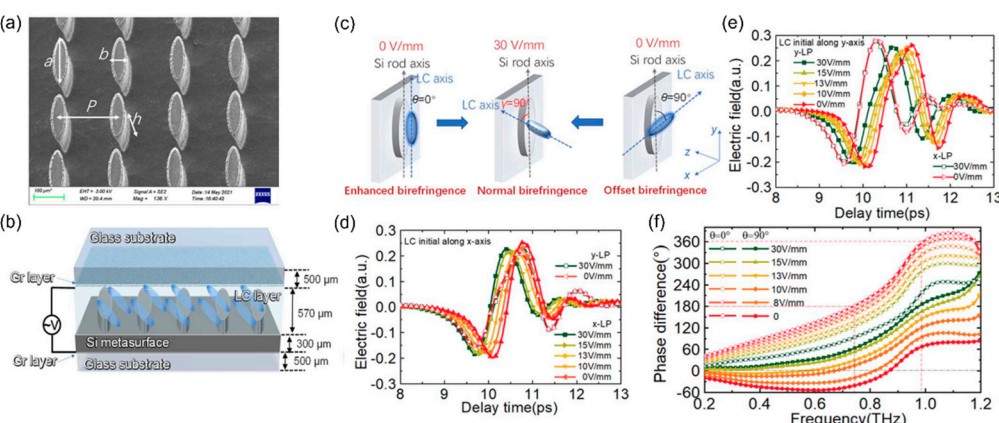

**Figure 8.** (**a**) SEM image of the Si dielectric metasurface. (**b**) Structural diagram of the LC-Si metasurface at different applied voltages. (**c**) Illustration of LC orientation in the LC-Si metasurface when the biased electric field $E$ = 0 V/mm. The LC is initially anchored along the $y$-axis ($\theta$ = 0°, $\gamma$ = 0°) and the $x$-axis ($\theta$ = 90°, $\gamma$ = 0°). When $E$ = 30 V/mm, the LC is turned to the $z$-axis ($\gamma$ = 90°), and the long axis of the Si column is fixed along the y-axis. The experimental THz time-domain signals for $x-$LP and $y-$LP components with a biased electric field in the range of 0 to 30 V/mm when the LC is initially along (**d**) the $x$-axis and (**e**) the $y$-axis. (**f**) The experimental phase shift of the device under different initial orientations and biased electric fields [105].

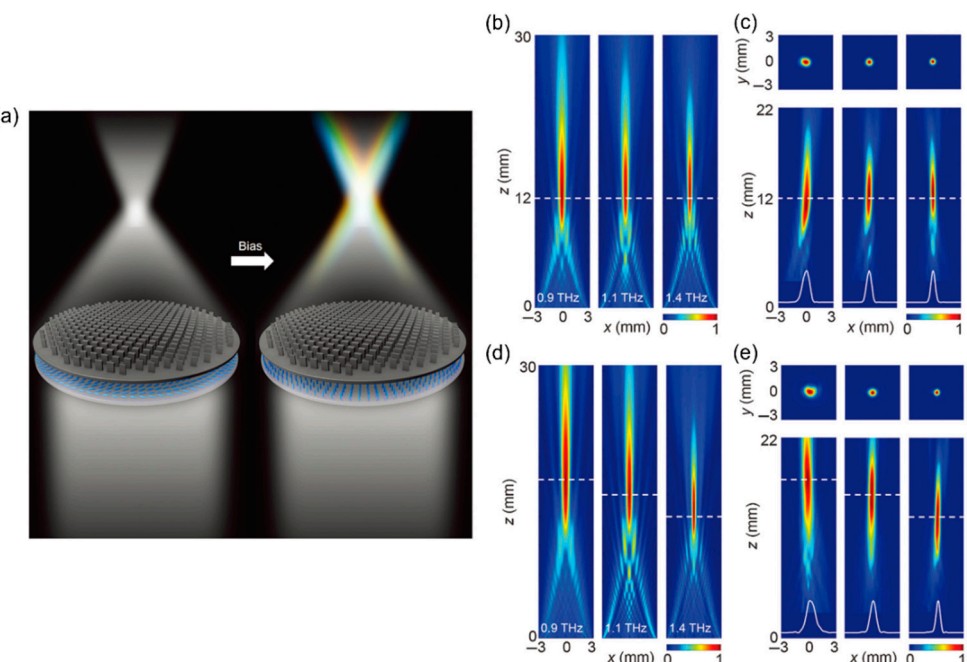

**Figure 9.** (**a**) The broadband achromatic focusing state without bias and the dispersive focusing state with a saturated bias. The superstrate depicts the dielectric metasurface, while the blue ellipsoids between the superstrate and substrate denote LCs. (**b**) Simulated THz fields in the *xz* plane and (**c**) measured THz fields in the *xz* and *xy* planes (z = 12.0 mm) at 0.9, 1.1, and 1.4 THz when no bias is applied. (**d**) Simulated and (**e**) measured THz fields in the *xz* and *xy* planes (z = 13.0 mm) of the same sample with a saturated bias [106].

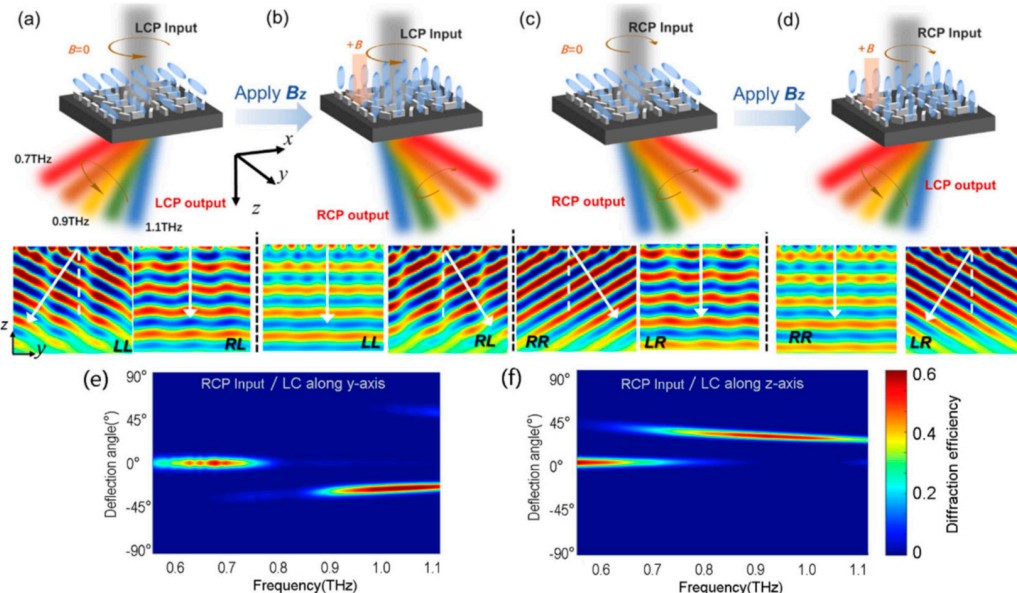

**Figure 10.** Schematic diagrams of the LC-PB metadevice in different cases: (**a**,**b**) LCP incidence; (**c**,**d**) RCP incidence. (**a**,**c**) LC molecules along the *y*-axis without an EMF; (**b**,**d**) LC molecules along the *z*-axis with an EMF. The simulated electric field distributions of the spin states at 1.0 THz are located below the corresponding diagram. The simulated diffraction efficiency map of the LC-PB metadevice varies with frequency and deflection angle when the RCP is incident: (**e**) LC molecules are along the *y*-axis and (**f**) *z*-axis [107].

However, the modulation speed has been rarely mentioned in these works on dielectric-LCs metadevices. The combination of LCs and dielectric metasurfaces supplies two distinguished ways to formdynamic functions: one case is that LCs are used to provide extra PB phase modulation, the typical thickness of which is hundreds of micrometers to meet the half-wave condition, and the modulation time is in second scale; the other is adopting gradient voltage to tune the birefringence in order to generate a gradual refractive index change with a millisecond scale response.

## 5. Prospect and Conclusions

Liquid crystals exhibit broadband birefringence and can be tuned by an external field, occupying a place in the modulation of THz waves. From THz filters and phase shifters to integrated devices, we can see the ubiquitous shadows of LCs. Their characteristic of external-field responsiveness enables the free manipulation of functions. Pure-LC-based THz modulators mostly rely on thick LC layers to accumulate certain phase retardation. For LC geometric phase devices, half-wave conditions need to be satisfied to maximize the efficiency. These requirements cause LC THz devices to suffer from slow responses. As a mainstream THz modulation technique, metasurfaces can modulate the amplitude, phase, polarization and many other dimensions of THz waves. Because the design does not rely on the traditional Snell's law and propagation phase accumulation, arbitrary functions can be realized with reduced thickness and compact size. By combining LCs with metasurfaces, a THz modulator combining LC modulation and metasurface modulation can be obtained. The compact, efficient and fast-response modulations of THz waves meet the development tendency toward multifunction, integration and miniaturization.

There are several aspects to be particularly pursued in future research. The first is the techniques of combining LCs and metasurfaces. For metal and dielectric metasurfaces, LCs are soaked on nanostructure arrays, as a refractive-index-tunable environment medium. For dielectric metasurfaces, LCs must be separated from the metasurface to provide PB phase modulation to avoid influences on the LC orientation of the nanostructures. Thus, new metadevice designs are needed to keep LCs separated from metasurfaces. The second is further improving the response time of LCs. The modulation speed of the THz modulator is a key requirement for most applications. The introduction of dual-frequency liquid crystals (DFLCs), blue-phase liquid crystals (BPLCs) and other fast response LCs may improve the modulation speed. The last is to introduce more kinds of external field. Most works in this review are electrically tuned. Their performance can be improved by further introducing other external fields, such as mechanical force, heat, light and magnetic fields. The responsiveness of LCs to multiple external fields enables the metasurface with free controllability. In addition to the innovations on LC materials and external-field excitations, the introduction of an artificial intelligence algorithm may facilitate many powerful application scenarios. The structure of a metasurface can be simplified by reverse design. The phase combination of both the LC and metasurface will inspire diverse THz photonic components. THz technology plays a key role in next-generation high-speed communication. Correspondingly, LC-integrated THz devices will also be extensively applied in communications.

**Author Contributions:** H.Y. wrote the draft of sessions 1 & 3, H.W. wrote the draft of session 2 and part of session 5, and Q.W. wrote the draft of session 4 and part of session 5. S.G., W.H. directed the preparation of the draft, and all authors participate in the preparation of the final manuscript. All authors have read and agreed to the published version of the manuscript.

**Funding:** The authors gratefully acknowledge the support of the National Key Research and Development Program of China (SQ2022YFA1200117), the National Natural Science Foundation of China (NSFC) (62035008, 62105143 and 61922038), Natural Science Foundation of Jiangsu Province (BK20210179) and Fundamental Research Funds for the Central Universities (021314380189).

**Conflicts of Interest:** The authors declare no conflict of interest.

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
