# Peer review of "Liquid Crystal-Tuned Planar Optics in Terahertz Range"

_applsci, doi:10.3390/app13031428_

Round 1

Reviewer 1 Report

Liquid crystals (LCs) exhibit broadband birefringence and can be tuned by an external field, and they are used to modulate THz waves. In this manuscript, the authors summarized the development of some LC THz devices such as tunable filters, absorbers, and programmable metasurfaces and anticipate the future developments of LC-integrated meta-devices in THz communication and imaging. The manuscript is comprehensive in content, reasonable in structure and standardized in writing. Please check the English expression carefully and it can be accepted after minor revision.

Author Response

Reviewer 1:

Liquid crystals (LCs) exhibit broadband birefringence and can be tuned by an external field, and they are used to modulate THz waves. In this manuscript, the authors summarized the development of some LC THz devices such as tunable filters, absorbers, and programmable metasurfaces and anticipate the future developments of LC-integrated meta-devices in THz communication and imaging. The manuscript is comprehensive in content, reasonable in structure and standardized in writing. Please check the English expression carefully and it can be accepted after minor revision. [Complied]

Thanks for the reviewer's comments. We have checked the English expression in the article and modified the “a zoom metalensin line 21 to “zoom metalens”, “provides in line 43 to “provide”, “Figure 1a and 1b” in line 105 to “Figures 1a and 1b”, “seen in Figure 2a” in line 135 to “as seen in Figure 2a”, “terahertz demultiplexers and spin controllable THz source” in lines 230-231 to “THz demultiplexer and spin controllable THz source”, “metamaterial” in line 250 to “metamaterials”. Indent lines 280, 289, and 293 by two characters. Modified the “which structure shown in Figure 8a and 8b” in line 337 to “whose structures are shown in Figures 8a and 8b”.

In addition to the English expression check, we also made changes to the content of the article. We correct the reference of Figure 1 (c) as reference 41 in line 113, revise line 16 “birefringences” to “birefringence”, line 342 “largest anisotropy” to “circular dichroism”, add “shown in Figure 1c,” in line 122, “which has a 47% modulation depth at 0.79 THz” in line 320 and “between 8.3 mm and 10.5 mm” in line 336. In order to make the manuscript more comprehensive, we add some descriptions in lines 202-204 as “Based on the rewritable photo-patterning of SD1, spatial terahertz modulator featured by photo-reconfigurable and electric-switchable is also designed.”

Reviewer 2 Report

In this work, dielectric meta-structure combined with LC was discussed comprehensively. The draft is helpful for colleagues. The work can be accepted for publication. However, the dynamic behavior such as modulation speed should be addressed. Therefore, the work can be accepted after including the dynamic issue. 

Author Response

Reviewer 2:

In this work, dielectric meta-structure combined with LC was discussed comprehensively. The draft is helpful for colleagues. The work can be accepted for publication. However, the dynamic behavior such as modulation speed should be addressed. Therefore, the work can be accepted after including the dynamic issue. [Complied]

The modulation speed has been rarely mentioned in the works on dielectric-LCs metadevices. As a compensation, we add corresponding discussions in part 4 in lines 378-383, “However, the modulation speed has been rarely mentioned in these works on dielectric-LCs metadevices. The combination of LCs and dielectric metasurfaces supplies two distinguished ways to form the dynamic functions: one case is that LCs are used to provide extra PB phase modulation, the typical thickness of which is hundreds of micrometers to meet the half-wave condition, and the modulation time is in second scale; the other is adopting gradient voltage to tune the birefringence in order to generate a gradual refractive index change with a millisecond scale response.”

Besides, we add some dynamic modulation state and parameter changes in part 4. In addition, we discuss some probable reason of the lack of modulation speed. We have revised our manuscript and added some descriptions in lines 325-327 as “This device can be turned from OFF state (no polarization conversion effect) to a controllable waveplate, exhibiting over 97% polarization conversion ratio (PCR) in 0.97-1.3 THz as a half-wave plate, or over 96% PCR in 0.67-1.3 THz as a quarter-wave plate”. Add descriptions of modulation depth and efficiency in lines 369-370 as “which realizes a dynamic modulation depth of >94% and maximum efficiency of over 50% for the different spin states”.

Reviewer 3 Report

In this manuscript, the birefringences of liquid crystals in the terahertz band and terahertz devices based on liquid crystals are summarized. And the future developments of LC-integrated meta-devices was discussed, which will inspire broad applications in terahertz communication and imaging. The timeliness, the breadth and accuracy of the manuscript is within the scope of the journal. In my opinion, the research is valuable to be published if the authors address the following issues:

1. In order to improve the legibility of the manuscript, the authors are suggested not use the abbreviations in Abstract.

2. The novelties of the manuscript should be emphasized deeply in Introduction.

3. As the author mentioned in Section 4, the function of the dielectric metasurface is strictly bound to the structure. Does the manufacture of the dielectric metasurface limit the development of their capabilities? The authors are suggested a brief description of this aspect.

4. The authors are suggested to indicate more detailed avenues for future research.

Author Response

Reviewer 3:

In this manuscript, the birefringence of liquid crystals in the terahertz band and terahertz devices based on liquid crystals are summarized. And the future developments of LC-integrated meta-devices was discussed, which will inspire broad applications in terahertz communication and imaging. The timeliness, the breadth and accuracy of the manuscript is within the scope of the journal. In my opinion, the research is valuable to be published if the authors address the following issues:

  1. In order to improve the legibility of the manuscript, the authors are suggested not use the abbreviations in Abstract.

[Complied]

We have changed the abbreviations “RF” to “radio frequencyin line 10, and “LC” to “liquid-crystal in line 23.

  1. The novelties of the manuscript should be emphasized deeply in Introduction. [Complied]

Thanks for the reviewer's constructive suggestion. The LC-integrated THz devices have developed rapidly in recent years. The manuscript gives a systematic review on the recent progress of this field and anticipates the development towards intelligent, versatile, and powerful THz components.

Correspondingly, some descriptions are added in lines 67-71 as “With the aid of artificial intelligence algorithms, LC-integrated metadevices promote the dynamic control of THz beams and are diversely applied in THz radar detection and compressed sensing imaging. However, the lack of systematic review of the recent progress in this field hinders the rapid meeting the requirement of the LC-integrated THz devices.”.

  1. As the author mentioned in Section 4, the function of the dielectric metasurface is strictly bound to the structure. Does the manufacture of the dielectric metasurface limit the development of their capabilities? The authors are suggested a brief description of this aspect.

[Complied]

We add a comment on the manufacture of dielectric metasurface in lines 310-311 as “In the THz band, silicon-based metasurfaces are often fabricated by photolithography and deep reactive ion etching (DRIE). The manufacture is matured and available for mass production.”.

  1. The authors are suggested to indicate more detailed avenues for future research. [Complied]

Thank the reviewer for the constructive suggestion. In the fifth section, we prospect more developing trends of the LC-integrated metadevices. Some descriptions are added in lines 410-415 as “In addition to the innovations on LC materials and external-field excitations, the introduction of artificial intelligence algorithm facilitates many powerful application scenarios. The structure of metasurface can be simplified by reverse design. The phase combination of both LC and metasurface will inspire diverse THz photonic components. THz technology plays a key role in the next generation high-speed communication. Correspondingly, LC-integrated THz devices will also be extensive applied in communications.”.

Reviewer 4 Report

This review is about liquid crystal-tuned planar optics in terahertz range. It is very well written with good results. However, prior to publish some minor revision should be taken into account.

1. The author mentioned in the page 2 that “The birefringence, Δn, is a key parameter of the LC that determines the function of wavefront regulation..” Give the appropriate explanation for this statement.

2. Add more references about THz absorbers.

3. Author can also add study of birefringence of liquid crystals doped with nanoparticles for THz range.

Author Response

Reviewer 4:

This review is about liquid crystal-tuned planar optics in terahertz range. It is very well written with good results. However, prior to publish some minor revision should be taken into account.

  1. The author mentioned in the page 2 that “The birefringence, Δn, is a key parameter of the LC that determines the function of wavefront regulation.” Give the appropriate explanation for this statement.

[Complied]

Liquid crystal terahertz modulators work on the accumulated phase modulation, which are related to the birefringence and thickness of liquid crystal. Thick cells suffer from slow responses and poor alignments, thus large birefringence is high demand for such modulators. Accordingly, “For instance, LC THz modulators work on the accumulated phase modulations, which are related to the birefringence and thickness of LCs. Thick cells suffer from slow responses and poor alignments, thus large birefringence is highly demanded.” is added in lines 84-87.

  1. Add more references about THz absorbers.

[Complied]

We have added the literature on terahertz absorbers as references 63-65.

  1. Author can also add study of birefringence of liquid crystals doped with nanoparticles for THz range.

[Complied]

We really appreciate the reviewer’s constructive suggestion. Accordingly, we added the sentence “The nanoparticles doped E7 LC leads to a 10% increase in birefringence in 0.3-3 THz” in lines 92-93.